# SuperMerge: An Approach for Gradient-Based Model Merging

## Abstract

Large language models, such as ChatGPT (Josh Achiam, 2023), Claude (Anthropic, 2024), or LLaMA (Touvron et al., 2023), are gigantic, monolithic, and possess the superpower to simultaneously support thousands of tasks. However, high-throughput applications often prefer smaller task-specific models because of their lower latency and cost. One challenge of using task-specific models is the incremental need for solving newer tasks after the model is already deployed for existing tasks. A straightforward solution requires fine-tuning the model again for both existing and new tasks, which is computationally expensive and time-consuming. To address this issue, we propose a model merging based approach called SuperMerge. SuperMerge is a gradient-based method to systematically merge several fine-tuned models trained on existing and new tasks. SuperMerge is designed to be lightweight and fast, and the merged model achieves similar performance to fully fine-tuned models on all tasks. Furthermore, we proposed a hierarchical model merging strategy to reduce the peak space requirement without sacrificing the performance of the merged model. We experimentally demonstrate that SuperMerge outperforms existing model merging methods (Ilharco et al., 2022; Yu et al., 2024; Yadav et al., 2024; Yang et al., 2024) on common natural language processing and computer vision tasks.

## 1 Introduction

The rapid development of foundational models has led to numerous challenges and opportunities. Majority of the foundational models are *generative*, wherein they are capable of "generating" the output given an input. The input and output both can be free-form text, images, or multi-modal. This is a paradigm shift from the earlier predictive models that had rigid and task-specific inputs and outputs and were unable to generate free-form text or images. Generative capabilities enable foundational models to learn several tasks simultaneously and generalize to tasks unseen during training. This has lead to research in the topics of multi-task learning for several sub-disciplines of machine learning, such as computer vision (Chen et al., 2018; Misra et al., 2016), natural language processing (Collobert & Weston, 2008; Dong et al., 2015), and recommendation systems (Song et al., 2024; Ma et al., 2018).

In this context, it has become common to train large foundational models on thousands of tasks that could be performed with modest accuracy. During training these models gain the ability to generate a response based on a vast repository of public data and commonsense knowledge. However, there are situations where the model is not adequately trained on organization-specific tasks, whose data is not available in the public domain. In such cases, fine-tuning these models to perform a specific task (or subset of tasks) is often required. For example, a general model could be outstanding at summarizing text, but could be mediocre at summarizing organization-specific technical jargon.

**Fine-Tuning.** There are a few major classes of fine-tuning approaches. The first approach is called end-to-end fine-tuning. In this approach *all* existing model parameters are adjusted to minimize a loss function using mini-batch gradient descent and back-propagation over a fine-tuning data set. For tasks like classification, a new classification head is added to the model, such that it learns to classify on the fine-tuning data set. The parameters in the classification head are initialized randomly. However, for fine-tuning larger models the compute needed for end-to-end fine-tuning becomes prohibitively large and therefore methods known as parameter efficient fine-tuning (or PEFT) were

introduced. These methods only tune a sub-set of the parameters, while others are held fixed. This dramatically reduces the memory and compute required for fine-tuning and achieves acceptable performance on the fine-tuning tasks (Hu et al., 2021; Zhang et al., 2023; Liu et al., 2022; Dettmers et al., 2024).

One such popular PEFT approach is LoRA (Hu et al., 2022). In LoRA, each weight matrix in the model $W$ of size $n$-by-$m$ is represented as $W + UV^\top$ where $U$ and $V$ are low-rank and of size $n$-by-$r$ and $m$-by-$r$ and respectively. The original parameters $W$ do not participate in the fine-tuning process and only weights in $U$ and $V$ are tuned using gradient descent. The number of additional parameters that LoRA introduces are therefore $r(n + m)$. $r$ controls the number of new fine-tuning parameters, where typically $r < 5$. Thus, a tiny fraction of tunable parameters are used for learning from the fine-tuning data set. This drastically reduces memory and compute requirements.

Deployed fine-tuned models often require updates to extend their capabilities to newer tasks. For instance, a model fine-tuned for the text-to-SQL task could be extended to handle data processing in NumPy or Pandas for providing comprehensive tooling experience for data scientists. However, these extension efforts are non-trivial. One alternative is to start from the pre-trained model and fine-tune the model again on existing and new tasks. Even with PEFT, such repetitive fine-tuning process can be computationally expensive, especially when multiple rounds of task updates are required.

**Limitations of Ensemble Learning Techniques.** Utilizing ensemble learning is another approach for enabling new model capabilities (Caruana et al., 2004). However, foundational models are large and therefore ensembling them is often challenging. Additionally, creating an model ensemble entails obtaining predictions from all ensemble components. Given that foundational models have high inference cost, obtaining predictions from all ensemble components becomes slow and costly. Another approach here is to learn a "router" model that simply routes the input to the appropriate model for inference. However, such "routers" are models themselves and they have to be retrained frequently for maintaining performance. Alternatively, some approaches propose ensembling different checkpoints of the same model that are saved while training or fine-tuning (Garipov et al., 2018; Wortsman et al., 2022; Fort et al., 2019). These approaches often work when models are trained for the same task, but are relatively ineffective in combining models fine-tuned on different tasks.

**Model Merging.** Given these challenges, there has been an emergence of approaches that propose combining model trained on disparate tasks —instead of the predictions— to form a single model. These techniques are collectively referred to as *model merging methods*. As model merging methods output a single model the number of inference parameters remain unchanged, plus the new model is capable of performing existing and new tasks. Secondly, merging techniques are very efficient as compared to fine-tuning or even parameter efficient fine-tuning, making them an attractive alternative for enhancing task generalization performance of fine-tuned models.

Model merging techniques start from two or more models whose weights have been tuned on different data sets (or tasks). Their main objective is to merge these models with minimal impact to performance and with lower cost as compared to fine-tuning on all data sets from scratch. Model merging techniques typically work with models from identical architectures, but with either same or different initialization or starting points. Examples of the model merging techniques with the same initialization are DARE (Yu et al., 2024), Model Soups (Wortsman et al., 2022), and TIES (Yadav et al., 2024). Similarly, examples of techniques that work on different initialization are OT Fusion (Singh & Jaggi, 2020), Git-Rebasin (Ainsworth et al., 2023), and ZipIT (Stoica et al., 2023).

As we show in Section 4 and Fig. 2, existing model merging techniques such as DARE (Yu et al., 2024), TIES (Yadav et al., 2024) and Task-Arithmetic (Ilharco et al., 2022) are sensitive to scaling hyperparameters. We show that every model layer does not participate equally in fine-tuning: some layers receive larger weight updates than other layers. Plus, every model layer does not participate equally across tasks: some tasks generate larger updates for a layer than others. In an attempt to address these challenges, we propose a principled merging method called SUPERMERGE. SUPER-MERGE is a supervised model merging approach that learns the contribution of each layer using a tiny amount of validation data, instead of relying of uniform layer-agnostic scaling hyperparameters. SUPERMERGE adaptively merges at an appropriate granularity to deliver models that achieve state-of-the-art performance. SUPERMERGE achieves this performance while introducing a minuscule number of tunable merging parameters. We demonstrate that SUPERMERGE achieves an average accuracy improvement of 5.8% across all tasks, while the per-task improvement is upto 49.4% over

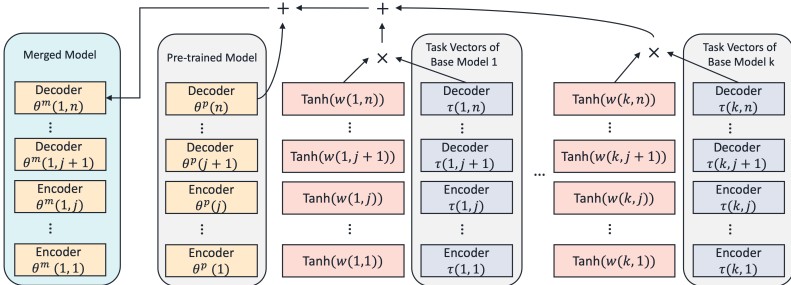

Figure 1: Overview of our proposed SUPERMERGE: The figure visualizes the merging steps of SUPERMERGE. The blue and yellow blocks represent the layer-wise task vectors and parameters in the large language model. The red blocks represent the layer-wise trainable parameters introduced by SUPERMERGE. For clarity, we only show the computation for the last layer of the merged model, as we presented in Eq. (1)

well-established and recent baseline techniques. In the nomenclature introduced earlier, SUPERME-RGE can be categorized as identical model and identical initialization approach.

## 2 RELATED WORK

**LLM Fine-Tuning Techniques.** The core idea of fine-tuning is to extend the knowledge acquired from pre-training and adapt it to a specific target domain through additional training using a task-specific data set. In the natural language processing domain, instruction fine-tuning (Wei et al., 2022; Chung et al., 2024) serves as a widely adopted approach to enhance the model's ability to comprehend and accurately execute desired tasks. To enhance fine-tuning efficiency, parameter-efficient fine-tuning (PEFT) methods have been proposed. PEFT methods typically introduce lightweight task-specific adaptations to the pre-trained model. One approach is to add adapters (Houlsby et al., 2019) as sub-modules to the pre-trained model, which enables efficient learning of new knowledge. Low-Rank Adaptation(LoRA) (Hu et al., 2022) factorizes each weight matrix into a low-rank decomposition that has minimal number of tunable parameters. Infused Adapter by Inhibiting and Amplifying Inner Activations (IA$^3$) (Liu et al., 2022) proposes a method of learning a subset of additional parameters to re-scale inner activations in the attention and feed-forward modules of transformer-based language models. These methods aim to achieve comparable performance to traditional full fine-tuning, while achieving significant reduction of tunable parameters.

**Multi-Task Learning via Model Merging.** Model merging methods aim to combine two or more fine-tuned models into a single model. Most existing methods focus on merging models that are derived from the same base architecture and initialization. The most intuitive approach is calculating a smooth average of the model parameters across different tasks, such as Task Arithmetic (Ilharco et al., 2022) and Fisher-Weighted averaging (Matena & Raffel, 2024). TIES-Merging (Yadav et al., 2024) takes a rule-based approach to resolve sign conflicts and merges with least feature redundancy. DARE (Yu et al., 2024) performs sparsification on each fine-tuned model by randomly dropping and ensembles them with rescaling. AdaMerging (Yang et al., 2024) is the most relevant related work to SUPERMERGE. Unlike SUPERMERGE, AdaMerging is an unsupervised approach that learns the layer contributions using an entropy-based loss function. This also restricts the ability of AdaMerging to only merging predictive models (classification and regression), while SUPERME-RGE can merge predictive and generative models. Another direction explores merging models with different initializations. The Git Re-Basin (Ainsworth et al., 2023) method permutes the units of one model to match them with the reference model, enabling the merging of two models in the aligned space. Optimal Transport Model Fusion (Singh & Jaggi, 2020) (or OT Fusion) leverages the Wasserstein barycenter to align weights or activations across different models, performing optimal transport computation in each layer to enable model fusion. Zipit (Stoica et al., 2023) merges models layer-by-layer by using a manually constructed computational graph. It uses this graph to examine the latent feature connections and redundancy that is identified through the similarity of

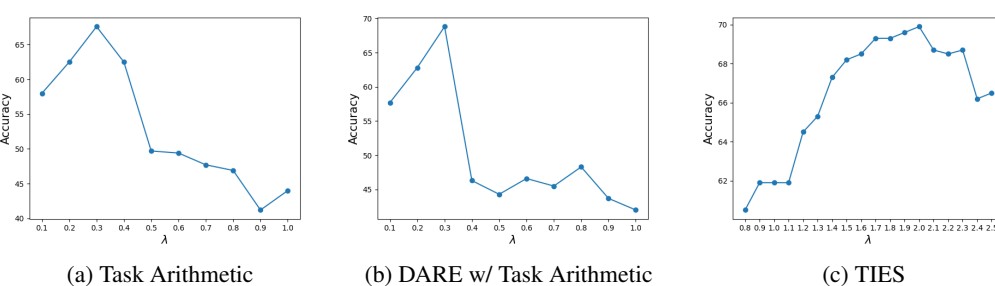

|     |     |     |
| :-: | :-: | :-: |
| (a) Task Arithmetic | (b) DARE w/ Task Arithmetic | (c) TIES |

Figure 2: Average performance of different model merging methods on 11 NLP tasks, using different hyperparameter $\lambda$.

activation values. However, it is challenging to apply Zipit to other models as such handcrafted computational graphs are not readily available.

## 3 NOTATION

The models that we consider for merging are deep learning models composed of numerous layers. Particularly, we assume that we have $k$ fine-tuned models with identical structure each having $n$ layers. We also assume that all the parameters of model $i$ are materialized into a single vector. We denote such a vector by $\boldsymbol{\theta}(i)$, where $i = (1 \ldots k)$. At times we need to refer to a particular layer $j = (1 \ldots n)$ of a fine-tuned model this is denoted as $\boldsymbol{\theta}^f(i, j)$. Next, we denote by $\boldsymbol{\theta}^p$ the weights of the pre-trained model and $\boldsymbol{\theta}^m$ denotes the weight of the final merged multi-task model. The fine-tuned and merged models have identical architecture as the pre-trained model.

A particular layer $j$ for pre-trained and merged model is denoted as $\boldsymbol{\theta}^p(j)$ and $\boldsymbol{\theta}^m(j)$ respectively[1]. We also assume that each of the fine-tuned models uses its own training, validation and test data sets. These data sets are denoted as $\mathcal{D}_1^{train} \ldots \mathcal{D}_k^{train}, \mathcal{D}_1^{val} \ldots \mathcal{D}_k^{val}$, and $\mathcal{D}_1^{test} \ldots \mathcal{D}_k^{test}$ respectively. We use $\mathcal{D}^{train}$, to indicate the union of all the training data sets. Similarly, $\mathcal{D}^{val}$ and $\mathcal{D}^{test}$ indicate the union of all the validation and test data sets. We assume that a fine-tuned model's parameters $\boldsymbol{\theta}^f(i)$ have been already tuned using their corresponding training data set $\mathcal{D}_i^{train}$.

## 4 BACKGROUND AND MOTIVATION

A majority of recent model merging approaches that follow the "identical architecture, identical initialization" merging style claim to be training-free[2]. In this category, there are three important and well-known model merging techniques relevant to SUPERMERGE: (1) Task-Arithmetic (Ilharco et al., 2022), 2) TIES (Yadav et al., 2024), and 3) DARE (Yu et al., 2024). Let us understand these methods with a toy setup of merging only two models, i.e. $k = 2$. The Task-Arithmetic approach begins by computing task vectors that are defined as $\boldsymbol{\tau}(1) = \boldsymbol{\theta}^f(1) - \boldsymbol{\theta}^p$ and $\boldsymbol{\tau}(2) = \boldsymbol{\theta}^f(2) - \boldsymbol{\theta}^p$. The weights of the merged multitask model are given as $\boldsymbol{\theta}^m = \boldsymbol{\theta}^p + \lambda(\boldsymbol{\tau}(1) + \boldsymbol{\tau}(2))$. In the TIES approach, the final weights are still computed similar to Task Arithmetic, but the task vectors $\boldsymbol{\tau}(1)$ and $\boldsymbol{\tau}(2)$ are computed differently. The TIES task vectors are sparsified by first keeping only the top-$k$ entries. Then, positive and negative values of a particular parameter across models are independently averaged. The final parameter value is the one with larger average magnitude. In the DARE approach, the task vectors of each model $\boldsymbol{\tau}(1)$ and $\boldsymbol{\tau}(2)$ are first sparsified by randomly setting the parameters in $\boldsymbol{\tau}(i)$ to zero with probability $p$. The remaining parameters are then rescaled by multiplying them with $1/(1 - p)$ to form the modified task vectors $\tilde{\boldsymbol{\tau}}(i)$ and $\tilde{\boldsymbol{\tau}}(2)$. Rescaling is performed to regain the scale that existed before sparsification. The modified task vectors $\tilde{\boldsymbol{\tau}}(i)$ are combined using $\lambda$ to form the merged parameter vector $\boldsymbol{\theta}^m = \boldsymbol{\theta}^p + \lambda(\tilde{\boldsymbol{\tau}}(1) + \tilde{\boldsymbol{\tau}}(2))$.

---

[1]The model index $i$ is not required in this notation as there is a single pre-trained and merged model.
[2]These approaches still need data to estimate hyperparameter values.

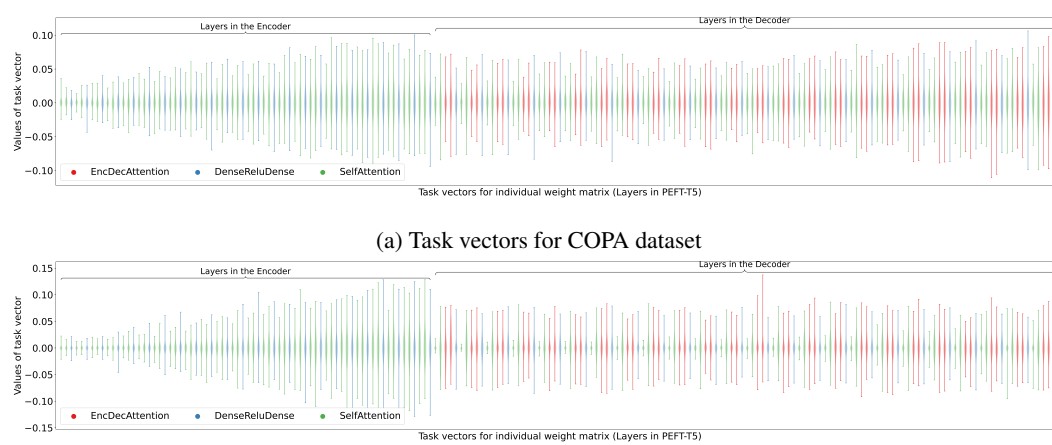

(a) Task vectors for COPA dataset

(b) Task vectors for RTE dataset

Figure 3: Variation of task vectors in PEFT fine-tuned model T5-IA[3]. Each violin plot along the $x$-axis visualizes the distribution of the task vector per weight matrix (per layer). The $y$-axis shows the magnitude of task vector. The first half of the figure (blue and green) plots the layers in the encoder and the second half (blue, green, and red) plots the layers in the decoder. Observe that different layers in the same models have different magnitudes, and there is a clear cut-off between encoder and decoder. Similar pattern is observed for same layer in different models. For example, the first few layers of encoder (left most ones) show smaller magnitude than the latter layers. We provide the full size figures in the appendix for better readability (Fig. 6a and Fig. 6b).

Observe that the existing so-called "training-free" model merging methods are not entirely devoid of training procedures. Crucially, the weighting parameter $\lambda$ involved in the Task Arithmetic, TIES-Merging, and DARE can be characterized as a form of hyperparameter, whose optimal value is found using the validation data set $\mathcal{D}^{val}$. These methods typically employ a single hyperparameter $\lambda$ to scale the task vectors and incorporate it into the pre-trained weight. $\lambda$ is determined through an extensive grid search process spanning values between 0 and 1, a non-trivial amount of computational resources are still required to identify this optimal value. As shown in Fig. 2, when we merge models across 11 different NLP tasks, the performance of DARE drop significantly on both sides of the optimal $\lambda = 0.3$. Similar behaviour is observed for TIES. Therefore, although the merging algorithm itself does not require any data, performing grid-search to the best $\lambda$ requires data and setting $\lambda$ without grid-search is detrimental to the merged model's performance (refer Fig. 2).

Next, we analyse the layer-wise distribution of the task vectors. For this, we PEFT tuned the T5 model Raffel et al. (2020) on two tasks, computed the layer-wise task vectors, and plot them in Fig. 3. Observe that the layer-wise distribution of task vectors could vary significantly across different layers and tasks. The distribution of the exhibits distinct patterns across different blocks of the T5 model. Both encoder and decoder demonstrate a general trend of increasing variance from initial to deeper layers. Additionally, the pattern of variation is task-specific: the same layer can receive different magnitude of updates on two different tasks. This observation demonstrates the limitations of employing a universal hyperparameter (such as $\lambda$) to scale the merged task vector, as different blocks and weight matrices exhibit distinct magnitudes of variations.

## 5 LEARNING TO MERGE MODELS

Given the challenges for selecting the optimal value for $\lambda$ and the large layer-specific variations in the task vectors, we propose a simple supervised approach for model merging called SUPERMERGE. SUPERMERGE introduces a tiny number of trainable parameters that can be tuned using a small amount of validation data $\mathcal{D}^{val}$. The number of parameters we introduce is proportional to the number of model layers. Typically, the number of layers is in the few hundreds, while the number of model parameters could be in the billions. As a result, we end up introducing only a minuscule number of new parameters.

Concretely, we assign a trainable weight to each layer $j$ of the fine-tuned model $i$. Let us denote such a weight by $w(i,j)$. In the SUPERMERGE approach, the weights of layer $j$ of the merged multi-task model are computed as:

$$\boldsymbol{\theta}^m(j) = \boldsymbol{\theta}^p(j) + \sum_{i=1}^{k} \tanh(w(i,j))\boldsymbol{\tau}(i,j) = \boldsymbol{\theta}^p(j) + \sum_{i=1}^{k} \tanh(w(i,j))(\boldsymbol{\theta}^f(i,j) - \boldsymbol{\theta}^p(i,j)) \tag{1}$$

Similarly all the layers are merged to form the final merged multi-task model $\boldsymbol{\theta}^m$. By adding $\tanh$ non-linearity we ensure that the merging weights can only wiggle between $(-1,1)$. As we will experimentally demonstrate in Section 6.2, this approach improves performance by avoiding over-fitting and improving generalization. All the weights $w(i,j)$ are tuned using gradient descent that minimizes loss on the validation data $\mathcal{D}^{val}$. The loss function is constructed as follows.

Assume that the trainable weights of the SUPERMERGE approach are collected in an $k$-by-$n$ matrix $\boldsymbol{W}$, where $\boldsymbol{W}_{i,j} = w(i,j)$ and denotes the trainable weight assigned to layer $j$ of model $i$. Let us denote by $f(\boldsymbol{x}; \boldsymbol{\theta}^m, \boldsymbol{W})$ the deep learning model with input $\boldsymbol{x}$ and parameters of the merged multi-task model $\boldsymbol{\theta}^m$ that is computed using the layer-wise merging approach given in Eq. (1) and the layer-wise weights $\boldsymbol{W}$. Then the loss that is minimized is given as follows:

$$\boldsymbol{W}^* = \arg\min_{\boldsymbol{W}} \mathbb{E}_{\boldsymbol{x},\boldsymbol{y} \sim \mathcal{D}^{val}} \ell(f(\boldsymbol{x}; \boldsymbol{\theta}^m, \boldsymbol{W}), \boldsymbol{y}), \tag{2}$$

where $\ell$ is the task-specific loss function, $\boldsymbol{x}$ and $\boldsymbol{y}$ are the input and output from the validation data. The intuition behind adding a tunable weight for each layer of each fine-tuned model is as follows. We have already seen that there is high degree of layer-wise variation of each task. Therefore, while forming the merged multi-task model it is important to learn the contribution of each fine-tuned layer to form the layer parameters of the merged model. Plus, we allow parameters $\boldsymbol{W}$ to vary between $(-1,1)$ by adding the $\tanh$ non-linearly. By allowing negative weights, we also learn which layers should be de-emphasized in addition to the layers whose contributions should be emphasized. Recall that $\boldsymbol{W}$ typically contains a few thousand entries that can be easily learned using a supervised approach. Although, $\boldsymbol{W}$ is difficult to learn using an unsupervised approach like AdaMerging Yang et al.. We experimentally demonstrate in Section 6 that using SUPERMERGE's gradient-based approach is superior than several existing approaches (including AdaMerging).

**Hierarchical Model Merging.** Observe that SUPERMERGE requires all the fine-tuned models to be in memory simultaneously for learning the weights $\boldsymbol{W}$. This memory requirement could be prohibitive when merging a large number of models. To address this issue, we devise a simple hierarchical strategy. The hierarchical approach limits the number of models that are simultaneously merged by forming a tree-like merging structure. In this structure the leaves are first merged to form intermediate models that are recursively merged in a breath-first manner to obtain the final merged model. The hierarchical merging process is in Fig. 4.

The hierarchical approach begins by merging models fine-tuned for similar tasks using SU-PERMERGE. This step combines closely related models first and preserves task-specific knowledge more effectively. In this step, we only use the validation data of the models being merged, further reducing memory requirement. We repeat the merging process using the intermediate models until we obtain a single merged model. In Section 6, we demonstrate that hierarchical SUPERMERGE is remarkably effective and memory efficient.

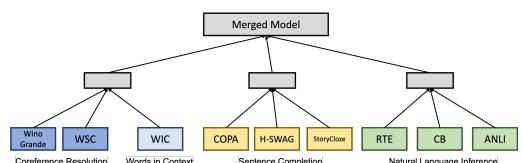

Figure 4: Illustration of hierarchical merging. Models from similar tasks (represented with a box of the same color) are merged to from intermediate models (represented with grey boxes).

## 6 EXPERIMENTAL EVALUATION

In this section we extensively evaluate SUPER-MERGE across multiple experimental settings. We test SUPERMERGE using both generative and

Table 1: In-domain performance comparison for generative tasks. In the table header, $N$ represents the size of the test data set. In the entries, the first number represents rank while the number in the parentheses represents accuracy.

| Rank $\downarrow$ (ACC $\uparrow$) | Avg. | rte (N=245) | cb (N=24) | winogrande (N=1235) | wic (N=606) | wsc (N=72) | copa (N=68) | h−swag (N=10010) | story_cloze (N=1839) | anli-r1 (N=1000) | anli-r2 (N=1000) | anli-r3 (N=1200) |
|---|---|---|---|---|---|---|---|---|---|---|---|---|
| Individual | 71.4 | 82.7 | 95.8 | 75.1 | 71.7 | 65.3 | 85.3 | 44.4 | 94.9 | 70.2 | 46.5 | 53 |
| Multitask | 73.1 | 88.6 | 95.8 | 75.5 | 61.1 | 80.6 | 94.1 | 42.3 | 97.6 | 70.5 | 49.8 | 47.7 |
| Task Arith. | 4.1 (63.3) | 5 (75.5) | 4 (79.2) | 2 (64.7) | 5 (49.7) | 4 (52.8) | 3 (89.7) | 4 (40.9) | 3 (95.3) | 5 (54.2) | 4 (47.0) | 6 (46.8) |
| DARE + Task Arith. | 4.5 (62.5) | 4 (75.9) | **1 (87.5)** | 5 (63.4) | 5 (49.7) | 5 (48.6) | 4 (88.2) | 5 (36.4) | 4 (94.3) | 6 (51.1) | 6 (45.1) | 5 (47.3) |
| TIES | 2.8 (65.8) | 3 (77.6) | 4 (79.2) | **1 (67.2)** | **1 (63.0)** | 3 (59.7) | 5 (86.8) | 3 (42.1) | 5 (89.3) | 2 (60.7) | 2 (48.2) | 2 (50.1) |
| DARE + TIES | 5.1 (57.0) | 6 (69.8) | 6 (75.0) | 6 (56.8) | 3 (55.1) | 5 (48.6) | 6 (69.1) | 6 (34.9) | 6 (66.0) | 3 (58.9) | 5 (45.5) | 4 (47.7) |
| SUPERMERGE | **2.2 (69.6)** | 2 (81.2) | 6 (75.0) | 3 (64.0) | 4 (54.5) | **1 (69.4)** | **1 (95.6)** | 2 (62.9) | 2 (96.0) | **1 (65.9)** | **1 (50.7)** | **1 (50.7)** |
| Hierarchical SUPERMERGE | 2.3 (69.4) | **1 (84.1)** | 2 (83.3) | 4 (63.5) | 2 (56.8) | 2 (62.5) | 2 (92.6) | **1 (71.0)** | **1 (96.2)** | 4 (56.9) | 3 (47.1) | 3 (49.0) |

predictive tasks. We purposefully choose generative tasks from the NLP domain and predictive tasks from the computer vision domain to demonstrate the broad range of tasks that are supported by SUPERMERGE. We have setup the experiments as follows.

**Generative Tasks.** We follow the setting used in (Liu et al., 2022; Yadav et al., 2024). Specifically, we use T0-3B (Sanh et al., 2022) as the base model and we use the IA$^3$ PEFT method (Liu et al., 2022) for fine-tuning. We fine-tune models on 11 data sets. Three sentence completion data sets (COPA (Roemmele et al., 2011), H-SWAG (Zellers et al., 2019), and Story Cloze (Sharma et al., 2018)). Three natural language inference (ANLI (Nie et al., 2019), CB (De Marneffe et al., 2019), and RTE (Dagan et al., 2005)). Two co-reference resolution data sets (WSC (Levesque et al., 2012) and Winogrande (Sakaguchi et al., 2021)) and a word sense disambiguation data set (WiC (Pilehvar & Camacho-Collados, 2018)). When fine-tuning IA$^3$ parameters added to the T0-3B model, the prompt are generated using templates from the Public Pool of Prompts (P3 (Bach et al., 2022)).

**Predictive Tasks.** We test SUPERMERGE on predictive tasks of image classification. For this, we use the Vit-B/32 architectures in CLIP (Radford et al., 2021) as our pre-trained models to conduct experiment on eight image classification datasets: Cars (Krause et al., 2013), DTD (Cimpoi et al., 2014), EuroSAT (Helber et al., 2019), GTSRB (Stallkamp et al., 2011), MNIST (LeCun et al., 1998), RESISC45 (Cheng et al., 2017), SUN397 (Xiao et al., 2016), and SVHN (Netzer et al., 2011). This setting is similar to the experimental setting used by (Ilharco et al., 2022; Yadav et al., 2024; Yang et al.). Some image classification data sets, like MNIST, don't include a validation set. In such cases, we use 10% of the training samples as the validation set.

**Baselines.** We use two types of baseline methods: non-gradient based methods and gradient based methods. *Non-gradient based* methods do not use gradient descent to adjust the tunable parameters. Here we use **Task Arithmetic** (Ilharco et al., 2022), **DARE** (Yu et al., 2024), and **TIES** (Yadav et al., 2024). These methods use the hyper-parameter $\lambda$ to add the merged task vector to pre-trained weight. *Gradient based* methods use gradient descent to tune the weights. Here we consider: **Individual** fine-tuned models that are fine-tuned using $D_i^{train}$. A single **Multitask** model fine-tuned using $\mathcal{D}^{train}$. Finally, we consider **AdaMerging**$^3$ (Yang et al.), which is a gradient-based model merging methods. It uses an unsupervised loss to learn the merging weight. We omit older baselines such as **Fisher Merging** (Matena & Raffel, 2024) and **RegMean** (Jin et al.) as they are known to be sub-optimal as compared to the recent baselines, such as TIES and DARE that are included in our comparison.

## 6.1 Performance of SUPERMERGE

In this section we analyse the in-domain and out-of-domain performance of SUPERMERGE. In-domain performance refers to the performance where the validation and test data set are sampled from the same distribution, otherwise it is considered out-of-domain performance.

**In-Domain Performance.** For both generative and predictive tasks, we use the validation set $\mathcal{D}^{val}$ to optimize the merging weights $\boldsymbol{W}$. A detailed analysis of the cost is presented in Section 6.2. The size of the validation set is much smaller than the training set. For example, the validation set for all

---

$^3$We use AdaMerging++ as our baseline as it has been shown to perform better than AdaMerging by (Yang et al.). For brevity, we refer to AdaMerging++ as AdaMerging.

Table 2: Out-of-domain performance comparison for generative tasks. In the table header, $N$ represents the size of the test data set. In the entries, the first number represents rank while the number in the parentheses represents accuracy.

| Rank ↓ (ACC ↑) | Avg. | cosmos_qa (N=2953) | social_iqa (N=1922) | paws (N=8000) | quail (N=556) | wiki_qa (N=6165) | quartz (N=784) | qasc (N=894) | ropes (N=1656) |
|---|---|---|---|---|---|---|---|---|---|
| Task Arith. | 3.1 (62.0) | 3 (64.6) | 3 (48.2) | 5 (70.3) | **1 (54.5)** | 5 (38.1) | **1 (71.9)** | 3 (90.8) | 4 (57.3) |
| DARE + Task Arith. | 3.3 (62.0) | 4 (63.1) | 4 (47.8) | 4 (70.8) | 2 (51.4) | 4 (43.6) | 2 (69.9) | 4 (90.7) | 2 (58.4) |
| TIES | 4.1 (59.2) | 5 (57.2) | 4 (47.8) | 3 (73.4) | 4 (50.0) | 3 (53.2) | 4 (65.9) | 5 (82.8) | 5 (43.3) |
| DARE + TIES | 6.0 (42.2) | 6 (28.3) | 6 (38.4) | 6 (64.6) | 6 (38.3) | 6 (34.8) | 6 (56.0) | 6 (58.3) | 6 (19.0) |
| SUPERMERGE | 2.1 (69.0) | **1 (68.5)** | 2 (48.2) | 2 (83.5) | 3 (50.9) | 2 (83.3) | 5 (65.1) | **1 (93.6)** | **1 (59.0)** |
| Hierarchical SUPERMERGE | **2.3 (69.1)** | 2 (67.2) | **1 (48.5)** | **1 (85.5)** | 5 (48.0) | **1 (84.7)** | 3 (68.8) | 2 (91.7) | 3 (58.1) |

Table 3: In-domain performance comparison for predictive task (image classification). In the table header, $N$ represents the size of the test data set. In the entries, the first number represents rank while the number in the parentheses represents accuracy.

| Rank ↓ (ACC ↑) | Avg. | SUN397 $N = 19887$ | Cars $N = 8041$ | RESISC45 $N = 6300$ | EuroSAT $N = 2700$ | SVHN $N = 26032$ | GTSRB $N = 12630$ | MNIST $N = 10000$ | DTD $N = 1800$ |
|---|---|---|---|---|---|---|---|---|---|
| Individual | 90.5 | 75.3 | 77.7 | 96.1 | 99.7 | 97.5 | 98.7 | 99.7 | 79.4 |
| Multitask | 88.9 | 74.4 | 77.9 | 98.2 | 98.9 | 99.5 | 93.9 | 72.9 | 95.8 |
| Task Arith. | 5.0 (70.1) | 5 (63.8) | 5 (62.1) | 5 (72.0) | 5 (77.6) | 5 (74.4) | 5 (65.1) | 5 (94.0) | 5 (52.2) |
| TIES | 3.9 (73.6) | 4 (64.8) | 4 (62.9) | 4 (74.3) | 4 (78.9) | 4 (83.1) | 4 (71.4) | 3 (97.6) | 4 (56.2) |
| AdaMerging | 2.9 (81.9) | 3 (66.1) | 3 (71.1) | 3 (80.2) | 3 (94.5) | 3 (83.8) | 3 (94.4) | 4 (97.4) | **1 (67.6)** |
| SUPERMERGE | **1.4 (85.7)** | **1 (67.9)** | **1 (73.4)** | **1 (90.7)** | 2 (97.6) | **1 (95.1)** | **1 (96.2)** | 2 (97.9) | 2 (66.4) |
| Hierarchical SUPERMERGE | 1.6 (85.6) | **1 (67.9)** | **1 (73.4)** | 2 (90.5) | **1 (98.3)** | 2 (94.0) | 2 (95.9) | **1 (98.5)** | 3 (66.2) |

generative NLP tasks consists of only 32 data points. The experimental results of in-domain tests for generative and predictive tasks are shown in Table 1 and Table 3. We report the performance for each individual task and the average performance across all tasks using two measures: accuracy and rank. Accuracy is a better choice when measuring performance on individual tasks. However, if averaged then easier tasks with higher accuracy could dominate the average. To mitigate this issue, we also report rank and average rank. Rank is more interpretable in real-world applications, making it easier to select the best model. As shown in Table 1 and Table 3, SUPERMERGE achieves the best performance in terms of both average accuracy and average rank. In the generative NLP experiment, SUPERMERGE shows particularly strong performance in sentence completion tasks like COPA, and H-SWAG, and Story_Cloze, where it ranks first with notable margins (6.5%, 68.8%, and 0.9%) to the second best model. Individually, SUPERMERGE ranks first in 8 out of 11 generative NLP tasks and in 5 out of 8 predictive CV tasks, showing strong performance across a wide collections of tasks.

**Out-of-Domain Generalization.** To further test the generalization performance, we evaluate merged models on out-of-domain or unseen data set and tasks. In the generative NLP experiment, we assess SUPERMERGE's performance on several hold out data sets. Particularly, we have included one paraphrase identification data set, PAWS (Zhang et al., 2019), and seven held-out Q&A data sets: ROPES (Lin et al., 2019), Cosmos QA (Huang et al., 2019), Social IQA (Sap et al., 2019), QuAIL (Rogers et al., 2020), QASC (Khot et al., 2020), WikiQA (Yang et al., 2015), and QuaRTz (Tafjord et al., 2019).

Table 2 shows that SUPERMERGE is robust when applied to out-of-domain tasks, achieving the highest average scores among baselines. While SUPERMERGE doesn't always have the highest score for every task, it consistently outperforms on all unseen tasks and never falls significantly behind the best model in any task. Another observation is that DARE+TIES performs poorly on these unseen data sets, with the lowest average accuracy of 42.2%. This may suggest that removing too many small values in the task vector reduces generalization performance, which is especially important for harder unseen tasks. For example, the accuracy score of DARE+TIES on the ROPES data set is only 19%, which is almost 30% lower than SUPERMERGE. TIES also suffers from poor generalization. In general, Task Arithmetic's out-of-domain performance is better than TIES, while TIES is better than Task Arithmetic in the in-domain test.

Unlike generative tasks, the out-of-domain generalization test for the predictive classification task is hard to evaluate since each base model has its own classification head for predictions. This changes the architecture of the fine-tuned model making it impossible to merge. This is a well-known issue for classification tasks in real-world applications. Thus, we do not include the out-

of-domain generalization test for the predictive classification tasks in this section. Although for completeness, we report the corresponding results in Table 6 included in the Appendix.

**Hierarchical Merging.** We analyse the performance of the SUPERMERGE's hierarchical model merging procedure in Tables 1, 2 and 3. Notably, hierarchical SUPERMERGE achieves similar performance to SUPERMERGE, in both in-domain and out-of-domain experiments. The comparable or improved performance is achieved while addressing the memory limitation of the standard SUPER-MERGE. As we will show in Section 6.2, hierarchical SUPERMERGE significantly reduces the peak memory requirement as it only requires a small subset of models in memory at any given instance.

## 6.2 ANALYSIS OF SUPERMERGE

**Restricting merging weights to (-1, 1).** We analyse the performance of SUPERMERGE with and without the use of the $\tanh$ non-linearity. As shown in Table 4, eliminating $\tanh$ leads to performance deterioration in both in-domain and out-of-domain experiments. Without using the $\tanh$ non-linearity, the magnitude of the merging weights are larger and a subset of fine-tuned models could become dominant in the merged model leading to poor generalization.

**Visualizing merging weights W.** The merging weights $W$ for the predictive task for SU-PERMERGE and AdaMerging are shown in Fig. 5a. Predictive tasks are used as, unlike SUPERMERGE, AdaMerging does not support generative tasks. The $x$-axis of Fig. 5a represents different layers of the ViT model, and the $y$-axis represents the weights of the 8 tasks introduced in Section 6. Each pixel represents the merging weight $w(i, j)$ for model $i$ and layer $j$.

Table 4: Ablation study on the activation $\tanh$. The models are evaluated using NLP tasks introduced in Section 6.

| Methods | In-Domain Average Accuracy | Out-of-Domain Average Accuracy |
|---|---|---|
| SUPERMERGE w/ tanh | 69.6 | 69.0 |
| SUPERMERGE w/o tanh | 64.8 | 67.1 |

AdaMerging only learns weights between $[0, 1]$ therefore it cannot learn to de-emphasize a particular layer's contribution, which is not the case for SUPERMERGE.

Another observation is that the merging weights learned by AdaMerging are sparse and merging only occurs in certain layers, e.g., the layers correspond to vertical blue strips. Thus, in AdaMerging many layers have no contribution to the final merged model. In contrast, the merging weights generated by SUPERMERGE are dense and move between $[-1, 1]$. Additionally, the merging weights in the earlier layers are sparser and closer to zero (lighter in color) compared to those in deeper layers. This merging weight distribution is compatible with image classification tasks, as earlier layers in vision models tend to capture low-level features that are not task-specific.

Next, we visualize the merging weights learned by SUPERMERGE for generative tasks in Fig. 5b. The $x$-axis represents different transformer blocks in the T5-IA[3] model, and the $y$-axis represents the 11 NLP tasks introduced in Section 6. We further group the merging weights according to the type of the corresponding task vectors. For instance, in the bottom right figure in Fig. 5b, we plot all the merging weights *w.r.t.* the feed forward networks (FFN) in the decoder, across different transformer blocks. This visualization suggests that different types of weight matrices in decoder and encoder have different distribution of merging weights. Most of the merging weights of the FFN layers in the decoder are positive, while weights of the self-attention layers are widely spread between $(-1, 1)$. This suggests a future research direction where weight-specific constraints are applied to the merging weights.

**Analysing Computational Cost.** Although SUPERMERGE is a gradient-based methods its cost is far lower than fine-tuning because SUPERMERGE tunes only a minuscule number of merging weights $W$. Also, SUPERMERGE only uses the validation set, instead of the larger training set, further reducing the computational cost. In Table 5, we provide an empirical analysis of the computational cost. SUPERMERGE displays a significant advantage with only 2,112 trainable parameters, substantially lower than the fully fine-tuned method (2.85B). FLOPs (Floating Point Operations) per epoch is calculated according to the number of parameters and data set size. Compared to fine-tuning methods, SUPERMERGE requires 1000x lower FLOPs. Fine-tuned models are trained on the training set, which consists of 254164 data points, while SUPERMERGE uses a much smaller

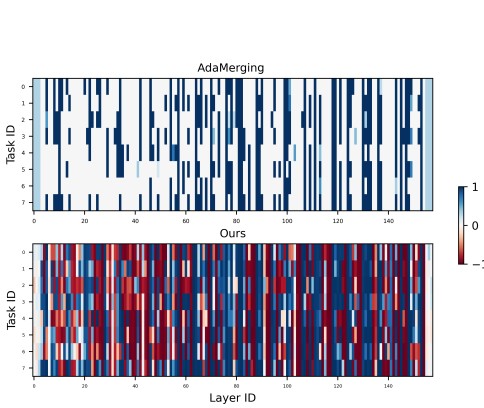

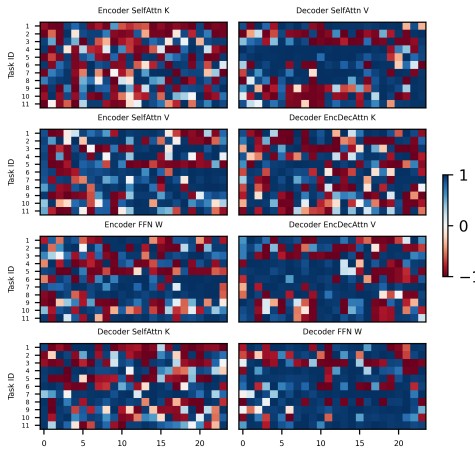

(a) Comparing merging weights $W$ for AdaMerging and SUPERMERGE for image classification.

(b) Visualizing merging weights $W$ learned by SUPERMERGE on generative tasks.

Figure 5: Visualization of merging weights.

Table 5: Computation cost and peak memory requirement of our methods and fine-tuning methods when merging 11 generative NLP tasks. The metrics are computed based on T5-3B model. More details are presented in Section 6.2

| Method | Number of Parameters | Number of Trainable Parameters | Peak Memory Requirement | Number of Training Samples | FLOPs per Epoch |
|---|---|---|---|---|---|
| Full Fine-Tuning | 2.85 B | 2.85 B | 43.5 G | 254164 | 3.6e16 |
| Non-Gradient Based Merging | 2.85 B | 1 | 130.4 G | 352 | 3.3e13 |
| SUPERMERGE | 2.85 B | 2112 | 130.4 G | 352 | 5.8e13 |
| Hierarchical | 2.85 B | 2112 | 32.7 G | 352 | 5.8e13 |

validation set consisting of 352 data points. Consequently, the reduction in trainable parameters and the use of a smaller data set allows SUPERMERGE to use orders of magnitude lower FLOPs as compared to fine-tuning.

**Analysing Peak Memory.** In Table 5, we report the peak memory requirement on generative NLP tasks where 11 T5-3B models were merged ($k = 11$). The exact details of this calculation can be found in Eq. (3) of Appendix A.2. As shown in the table, full fine-tuning has 2.85B trainable parameters and requires 43.5 GB of peak memory. Both non-gradient-based merging methods (e.g., TIES) and SUPERMERGE significantly reduce the number of trainable parameters compared to full fine-tuning (from 2.85B to 1 and 2112, respectively). However, SUPERMERGE and non-gradient based merging methods, such as Task Arithmetic, require much more peak memory than full fine-tuning, as these model merging methods need extra space to save task vectors for all the tasks. In contrast, SUPERMERGE's hierarchical merging approach merges as few as two models at a time, significantly reducing the peak space requirement from 130.4 GB to 32.7 GB.

# 7 CONCLUSION

We introduced a gradient-based model merging method called SUPERMERGE. SUPERMERGE only a handful of training parameters and runs significantly faster than full fine-tuning while maintaining similar performance. Our experiments demonstrate SUPERMERGE outperforms existing model merging methods on numerous tasks. Furthermore, merging several models at once could require substantial memory that increases with the number of tasks and the size of the base model. We proposed the hierarchical model merging strategy to alleviate this problem by merging a small subset of base models at a time. Hierarchical merging significantly reduces the peak memory requirement without compromising the merged model's performance.

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

## A APPENDIX

### A.1 ANALYSING OUT-OF-DOMAIN PERFORMANCE OF PREDICTIVE TASKS

In Table 6, we report the results of merging individual base models fine-tuned for tasks SUN397, Cars, SVHN, GTSRB, RESISC45, DTD, and the performance of the merged model on two unseen datasets, EuroSAT and MNIST, using the corresponding classification heads that are not merged. Unsurprisingly, all the comparison methods show significant performance drops on the unseen data sets. The performance of gradient-based methods, i.e., AdaMerging and SUPERMERGE, decreases more than non-gradient-based methods like TIES and Task Arithmetic, although recall that the gradient-based methods outperform the non-gradient-based methods in the in-domain task.

Table 6: Out-of-domain performance comparison for predictive task (image classification). In the table header, $N$ represents the size of the test data set. In the entries, the first number represents rank while the number in the parentheses represents accuracy.

| | | In-Domain | | | | | | Out-of-Domain | | |
|---|---|---|---|---|---|---|---|---|---|---|
| Rank ↓ (ACC ↑) | Avg. | SUN397 $N = 19887$ | Cars $N = 8041$ | SVHN $N = 26032$ | GTSRB $N = 12630$ | RESISC45 $N = 6300$ | DTD $N = 1800$ | Avg. | EuroSAT $N = 2700$ | MNIST $N = 10000$ |
| Individual | 87.4 | 75.3 | 77.7 | 97.5 | 98.7 | 96.1 | 79.4 | 99.7 | 99.7 | 99.7 |
| Task Arith. | 3.2 (62.8) | 4 (54.9) | 4 (55.1) | 3 (80.2) | 4 (69.7) | 3 (66.7) | **1 (77.2)** | **2 (87.3)** | 3 (50.1) | **1 (97.3)** |
| TIES | 2.8 (62.8) | 3 (60.9) | 3 (67.1) | 4 (75.6) | 3 (70.6) | 4 (60.7) | 4 (41.9) | 2 (74.9) | **1 (74.8)** | 3 (74.9) |
| AdaMerging | 2.0 (79.9) | 2 (65.6) | 2 (70.3) | 2 (88.4) | **1 (95.6)** | 2 (88.2) | 3 (71.3) | 2 (73.6) | 2 (61.4) | 2 (85.8) |
| SUPERMERGE | **1.3 (82.7)** | **1 (71.0)** | **1 (72.3)** | **1 (93.5)** | 2 (94.7) | **1 (90.9)** | 2 (73.9) | 4 (35.5) | 4 (39.0) | 4 (70.6) |

### A.2 CALCULATING PEAK MEMORY

The peak memory requirement is computed using the following equation:

$$\text{Peak Mem.} = \underbrace{4 * N_{\text{Para}}}_{\text{Model Weights}} + \underbrace{4 * N_{\text{Trainable Para}}}_{\text{Gradients}} + \underbrace{8 * N_{\text{Trainable Para}}}_{\text{Optimizer States}} + \underbrace{\mathbf{1}_{\text{merging}} * 4 * k N_{\text{Task Vector}}}_{\text{Extra Space for Task Vectors}} \quad (3)$$

where $k$ is the number of task. The peak memory requirement consists of four components. Memory required for all the weights of the pre-trained model. Each parameter needs 4 bytes space. Optimization requires extra space to save the gradients and optimizer states of trainable parameters. We chose AdamW as the optimizer and it needs 8 bytes per parameters to maintain 2 states. The space for task vectors of all $k$ task-specific fine-tuned models. Note that only model merging methods need extra space for the task vectors and fine-tuning approaches don't need to save any task vector since these methods use the pre-trained model as the starting point. Thus, we added a indicator function to the forth term in Eq. (3).

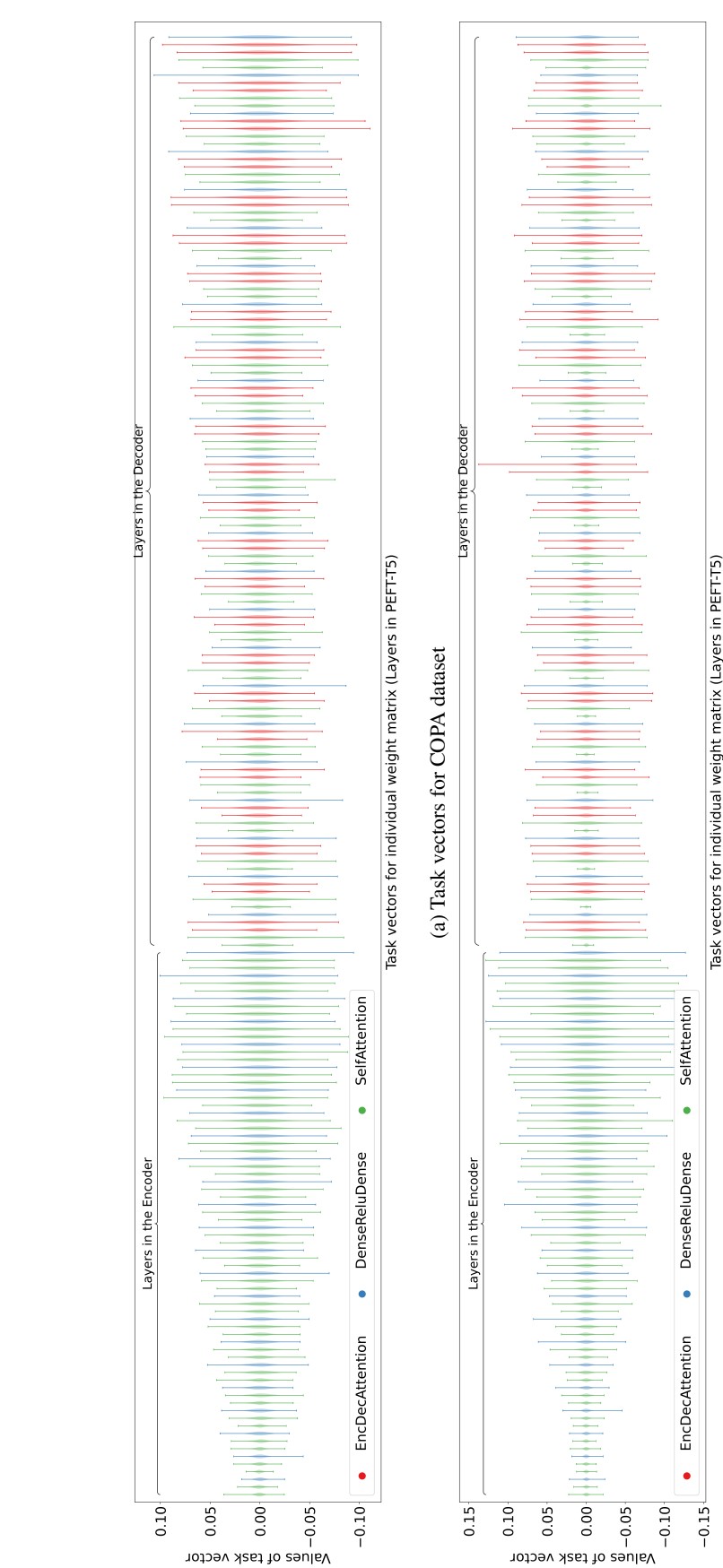

Figure 6: Variation of task vectors in PEFT fine-tuned model T5-IA$^3$. Each violin plot along the $x$-axis visualizes the distribution task vector per weight matrix. The $y$-axis shows the magnitude of task vector. Observe that different layers in the same models have different magnitudes. Similar pattern is observed for same layer in different models.

