# OpenReview forum: "SUPERMERGE: An Approach For Gradient-Based Model Merging"
_ICLR.cc/2025/Conference — ICLR 2025 Conference Withdrawn Submission_

### Official Review · Reviewer_ttUv · 2024-10-31

**Soundness:** 3
**Presentation:** 2
**Contribution:** 3
**Rating:** 5
**Confidence:** 3

**Summary:**

The paper **SUPERMERGE** presents a gradient-based model merging method to efficiently combine task-specific models without repeated fine-tuning. SUPERMERGE assigns adaptive weights to each layer, learning the best merge configuration using a small validation set. It also introduces a hierarchical strategy to reduce memory use when merging large models. The method outperforms existing approaches in accuracy across NLP and vision tasks and generalizes well to new data.

**Strengths:**

**Efficient Model Merging**: SUPERMERGE enables fast, gradient-based merging of task-specific models, avoiding repetitive and costly fine-tuning.

**Enhanced Performance**: SUPERMERGE demonstrates superior accuracy over other model-merging methods across NLP and computer vision tasks.

**Robust Generalization**: It performs well on out-of-domain data, showing adaptability to unseen tasks.

**Minimal Computational Overhead**: The method requires significantly fewer parameters and computational resources than full fine-tuning, improving efficiency.

**Weaknesses:**

I contend that there are the following weaknesses:

1. The paper should focus more on discussing the practical applications of this model merging method that requires no additional training and describe the specific advantages of this approach compared to multi-task learning. It seems that model merging may consume a large amount of GPU memory, especially for large language models (LLMs). If one aims to merge multiple LLMs, the memory usage could be extremely high.

2. Currently, LLMs typically use parameter-efficient fine-tuning to adapt to specific tasks, which allows merging only a minimal amount of LoRA parameters to achieve non-gradient model merging, such as in LoRAHub[1]. I believe this is more practically meaningful. Can your method be adapted to merge LoRA modules?

[1] LoraHub: Efficient Cross-Task Generalization via Dynamic LoRA Composition

**Questions:**

1. I think there should be a trade-off of memory and training time between SUPERMERGE and Hierarchical models. If it is right, can you provide the training time for the two methods?

2. With the introduction of instruction tuning, we only need to extract a small number of examples from each task for training to achieve performance similar to full fine-tuning, as demonstrated by models like Flan-T5 [2]. Therefore, if I have 10 tasks and select 1,000 examples from each for multi-task learning or instruction tuning, could this approach achieve better performance without significantly lagging behind the model merging method in terms of training time? I’m not entirely sure, so could the authors discuss this possibility?

[2] The Flan Collection: Designing Data and Methods for Effective Instruction Tuning

**Details Of Ethics Concerns:**

No Ethics Concerns

---

### Official Review · Reviewer_GWnA · 2024-11-02

**Soundness:** 1
**Presentation:** 1
**Contribution:** 2
**Rating:** 3
**Confidence:** 3

**Summary:**

This paper presents a new way to merge task-specific models in order to solve new tasks. The method is gradient-based trained on small validation sets. Task specific models are obtained by fine-tuning pre-trained models on task-specific data. SuperMerge is lightweight and fast, and it outperforms existing model merging methods on NLP and CV tasks.

**Strengths:**

1. SuperMerge is a gradient-based model merging method that can generalize to out of domain datasets.
2. The number of merged models are larger than previous works.
2. The authors also propose a hierarchical merging method that can reduce memory footage, with a slight performance decrease.

**Weaknesses:**

I recommend that the authors reorganize their paper, as the current version is difficult to follow. The current presentation of the paper does not meet the standard of ICLR in general. The major concern is that the experimental results cannot support the claims. In addition, I'm afraid that I have to disagree with some general statements that the authors made in the Introduction, see Questions point 1 in the following section.
1. **Reorganization of Presentation:**
    1. In Section 4, it would be beneficial to include the background information in the Related Work or Preliminary sections and to incorporate the motivation into the Introduction.
    2. The current Introduction contains several paragraphs that could be more appropriately placed in the Related Work section.
    3. numerous typos in the sentences and references need correction. See Questions section below.
2. **Experimental results:** The experimental results do not adequately support the claims made. The challenge identified by the authors is "incremental need for solving newer tasks after the model is already deployed for existing tasks" (line 16), which suggests a continual learning setting. However, the authors only conducted out-of-domain tests to evaluate the merged model's ability on unseen datasets, which does not constitute an incremental continual setting.
3. **Motivation for Model Merging:** The motivation for merging different task-specific models is not well articulated. The authors state in line 20 that "SuperMerge is ... a method to systematically merge several fine-tuned models trained on existing and new tasks." What is the purpose of merging these models if fine-tuned specific models already exist for these tasks? Especially considering that the results show merging the models does not improve accuracy compared to task-specific models, as seen in Tables 1 and 3.
4. **Clarification on Experimental Setting:** Following the previous comment, does SuperMerge also merge models trained on new tasks? This contradicts the authors' claim in line 413, where they "evaluate merged models on out-of-domain or unseen datasets and tasks." It is unclear in what scenario SuperMerge would merge models trained on new tasks. Can the authors provide an example scenario illustrating when and how SuperMerge would be applied to merge models for both existing and new tasks?
5. **Task Selection:** Why do the authors only conduct generative tasks for the NLP domain and predictive tasks for the CV domain? It is also possible to conduct discriminative tasks for the NLP domain and generative tasks for the CV domain.
6. **Comparison of Computational Cost:** When discussing computational cost, the authors emphasize the comparison with the fine-tuning methods. However, it would be more interesting to put more weights on the comparison with previous model merging methods. Furthermore, an additional possible baseline could be using gradient descent on the validation set to find the hyperparameter $\lambda$ used in previous model merging methods.
7. **Word Choices:** There are many strong words when discussing experimental results, for example, the term "significant-ly" appears 11 times in the paper. I suggest that the authors use these adjectives with caution to ensure that the language accurately reflects the results and maintains a balanced tone.

**Questions:**

1. **Questions related to the Introduction:**
    1. In line 35, the statement "Generative capabilities enable foundational models to learn several tasks simultaneously and generalize to tasks unseen during training" would benefit from references to support this claim. It is unclear why the generalization ability of foundational models is attributed to generative capabilities, and how the ability to learn several tasks simultaneously is related to these generative capabilities. I believe that multi-task learning is originally for discriminative tasks instead of generative tasks.
    2. In line 41, the claim "It has become common to train large foundational models on thousands of tasks ..." should be supported with references. To my knowledge, large foundational models are typically trained on a large corpus using the next token prediction task, rather than trained on thousands of tasks.
    3. In line 46, the statement "A general model could be outstanding at summarizing text, but could be mediocre at summarizing organization-specific technical jargon" seems to imply that technical jargon are not texts. It might be more accurate to use "general texts or open-domain texts" instead of only using "text".
    4. It is unclear why fine-tuning and ensemble learning are discussed in detail in the Introduction, as they do not seem directly related to the rest of the paper.
    5. In line 84, when introducting Model Merging, it would be helpful to include references that define it. For instance, DARE defines model merging as "to fuse the parameters of K fine-tuned models into a single model that can well handle K tasks simultaneously", without aiming to solve new tasks.
2. In Table 4, it would be interesting to see the performance for each task individually, in addition to the average accuracy.
3. There is an inconsistency in the number of data points used for SuperMerge. Line 400 mentions 32 data points, while line 516 states 354 data points. Clarification is needed.
4. Figure 2 indicates that previous models are sensitive to the hyperparameter $\lambda$. It would be interesting to explore whether SuperMerge is sensitive to the weight $w$.

**Typos:**
1. Line 36, has lead $\to$ has led
2. Line 74, an model $\to$ a model
3. Line 107, upto $\to$ up to
4. Line 298, Yang et al. add parentheses and publication year
5. Line 363, Jin et al. add publication year

---

### Official Review · Reviewer_ada1 · 2024-11-03

**Soundness:** 3
**Presentation:** 3
**Contribution:** 2
**Rating:** 5
**Confidence:** 4

**Summary:**

This paper introduces a method which learns individual scaling coefficients (defined in this paper by $\lambda$) for every layer, when merging models of the same architecture & initialization. The authors motivate their contributions by showing (1) optimal scaling coefficients are difficult to find in IA3 PEFT models, and (2) the magnitudes of task vector elements change across layers for the same IA3 models. Based on these observations, the authors propose to learn a distinct scaling coefficient for every layer of every model involved in merging. In an effort to mitigate computational expense, the authors further propose a hierarchical merging strategy, which iteratively learns the scaling coefficients for distinct model subsets, before the resulting merged models are themselves merged by learning new scaling coefficients. The paper achieves significantly better performance than cited baselines across all settings evaluated, while requiring only ~2x FLOPs.

**Strengths:**

1. The performance of SuperMerge is quite strong compared to the cited works, across all settings.
2. SuperMerge is an efficient learning framework, that achieves strong performance from few labeled examples.
3. The authors motivate their approach decently, though experimental results are the more compelling motivation.
4. The authors ablate SuperMerge well, showing that tanh delivers strong performance results and enables more flexible merges compared to existing work.
5. With the exception of a few places (mentioned below), the paper is written very clearly and well.

**Weaknesses:**

**Weaknesses**
1. **Insufficient baseline comparison:** Given that the authors propose a gradient-based model merging approach, I believe MaTS (https://arxiv.org/pdf/2312.04339) should be compared against. If I understand the settings between SuperMerge and MaTS correctly, MaTS appears to perform *dramatically better* than SuperMerge in the IA3 setting. I would imagine that SuperMerge is significantly more computationally efficient than MaTS, though it is difficult to tell given that the reported FLOP counts in SuperMerge and MaTS (see section 6.5 of their paper) are vastly different.
2. **What are the Total FLOPS?** How many FLOPS *total* does SuperMerge require? "FLOPs per epoch" is a bit misleading, as if I understand correctly, non-gradient based merging is a one time cost of 3.3e13 FLOPs, while the other three methods scale according to epochs? Total FLOPs is a much cleaner metric. This would also enable us to better understand the tradeoffs between computational expense and performance.
3. **I think the explanation in lines 423-424 may not be correct.** As far as I know, DARE+TIES can select to not prune any value based on the hyper-parameter search right? To me, the fault may be better explained by the sign conflict resolution in TIES. Specifically, perhaps the task vectors lie in high conflict with one another, so a significant portion of each model is pruned under conflict resolution?


**Writing Suggestions**
1. The introduction is severely bloated and reads like a related work section. In particular, I think paragraphs 3-4 should be merged with the first paragraph in the related work. Similarly, the entire section on model merging better belongs in the related work section. I would highly recommend the authors cut down on the introduction, and move the aforementioned paragraphs into the related work section. It would make the paper read substantially better as well.
2. Line 81 (last sentence of paragraph) needs a citation.
3. The definition of "Rank" should be introduced much earlier. I was very confused what "rank" meant as I was reading the paper. I think it is better to introduce this when showing results for the first time (e.g., caption of table 1).

**Questions:**

1. **Why does SuperMerge outperform Fisher and RegMean?** Fisher learns per-parameter scaling coefficients between models to merge, making it similar/relevant to SuperMerge (Both learn fine-grained scaling coefficients). Same goes for RegMean, which also learns fine-grained coefficients between models. Why do the authors believe SuperMerge is so much better?
2. **Confusion with Table 1 models:** I'm a bit confused by the models used in Table 1. Are these IA3 models used by the authors the same as those used in the TIES paper (https://arxiv.org/pdf/2306.01708)? The individual and multitask performances are equivalent between SuperMerge and TIES. However, the reported Task Arithmetic and TIES "Avg." results differ from what is reported in the TIES paper. If the models are different, I would ask that Fisher is used an additional baseline in this setting, as it is very relevant to the author's approach (otherwise it can be omitted it is shown to perform worse than TIES in the TIES paper). I doubt this will change the message of SuperMerge though, so including it as a question and not a weakness.
3. **I don't understand the comparison to the full-finetuned method in Table 5.** Shouldn't the result be "0" for FLOP count? SuperMerge (and all other merge methods) rely on the finetuned model to exist, so the comparison in Table 5 doesn't really make sense to me. I think it's better to remove the "Full-finetuning" method from this section entirely (it's not a fair comparison).


**Note to Authors:** I am willing to revise my score according to the rebuttal.

---

### Note · Authors · 2024-11-15

I have read and agree with the venue's withdrawal policy on behalf of myself and my co-authors.